# The Change in Habitat Quality for the Yunnan Snub-Nosed Monkey from 1975 to 2022

**DOI:** 10.3390/biology12060886

**Published:** 2023-06-20

**Authors:** Shuxian Zhu, Li Li, Timothy J. Slate, Haixia Tang, Gongsheng Wu, Hongyan Guo, Dayong Li

**Affiliations:** 1Key Laboratory of Southwest China Wildlife Resources Conservation (Ministry of Education), China West Normal University, Nanchong 637009, China; sxianzhu@126.com (S.Z.); tim.slate.vt@gmail.com (T.J.S.); 13072882907@163.com (H.T.); 2Key Laboratory of Conservation Biology of *Rhinopithecus roxellana*, China West Normal University of Sichuan Provence, Nanchong 637009, China; 3Land Improvement Center of Heping County, Heyuan 517200, China; 4Wildlife Management and Ecosystem Health Center, Yunnan University of Finance and Economics, Kunming 650221, China; wugongsheng@ynufe.edu.cn (G.W.); guohongyan2010@163.com (H.G.)

**Keywords:** Yunnan snub-nosed monkey, dynamic analysis, habitat quality, habitat degradation, habitat rarity, distance threshold value

## Abstract

**Simple Summary:**

The Yunnan snub-nosed monkey is one of the most endangered species on the IUCN Red List. The study of its population changes and its habitat quality and rarity changes over time are important for identifying opportunities for conserving and improving habitat quality while minimizing the adverse impact on ongoing human socio-economic development. Conserving and improving habitat quality is an important part of protecting and improving habitat diversity and biodiversity. Minimizing adverse impact to human development is important for earning energetic buy-in from the local people needed for long-term sustainability of such opportunities. Habitat Quality measures the land cover for an area at a given point in time compared to the preferences of a species or group of species. A high habitat quality indicates a large part of the area has land cover the species prefers, such as Huashan Pine for the Yunnan snub-nosed monkey. Habitat Rarity represents the change in rarity of a land coverage type (such as Huashan Pine) in an area over two points in time. A high habitat rarity indicates the land coverage type significantly decreased between the two points in time. A high Huashan Pine rarity, for example, means a significant decrease in the Yunnan snub-nosed monkey’s most preferred land cover and hence will likely significantly impact the monkey’s survival. Our analysis of the change in habitat quality and habitat rarity over time indicate increases in socio-economic developments in the villages around the habitat area have adversely affected the habitat quality and habitat rarity. This has resulted in a decline in biodiversity persistence, resilience, and breadth. It also has exacerbated the risk of declining species populations, potentially to extinction. Though focused on the Yunnan snub-nosed monkey, our approach toward the assessment of habitat quality and rarity over time based on species habitat suitability introduces a new perspective for assessing village development impacts on habitats for the conservation of other species.

**Abstract:**

The reduction in habitat quality (as shown, in part, by the increase in habitat rarity) is an important challenge when protecting the Yunnan snub-nosed monkey. We used the InVEST model to quantitatively analyze the dynamic changes in the habitat of the Yunnan snub-nosed monkey from 1975 to 2022. The results show that in the study period, the degree of habitat degradation increased, with the degradation range at its widest in the south, and the degradation intensity highest in the north, especially along a center “spine” area in the north. Over the latter part of the study period, the habitat quality of most monkey groups improved, which is conducive to the survival and reproduction of the population. However, the habitat quality and monkey populations are still at significant risk. The results provide the basis for formulating the protection of the Yunnan snub-nosed monkey and provide research cases for the protection of other endangered species.

## 1. Introduction

Climate change, changes in land use patterns, and human population expansion will lead to continued fragmentation of primate habitats and decline in habitat quality [1,2,3,4]. This increase in habitat fragmentation will lead to the isolation of various primate populations, resulting in reduced genetic diversity of the population [1,2,3,4]. Hence, even if the quality of an isolated habitat area is improved, the primate population may still decrease due to this reduced genetic diversity [5,6,7]. The quality and fragmentation of a habitat area is a useful measure for evaluating the viability of a primate species because of the tight dependence of the species on the habitat [8,9]. Hence, the evaluation of habitat quality and fragmentation, and deriving solutions from that evaluation to improve both, is critical for the protection of monkeys and biodiversity in a region [10,11,12].

The Yunnan snub-nosed monkey is an endangered primate species unique to southwest China, which is distributed in a narrow area between the Lancang and Jinsha rivers in the middle of the Yunling Mountains, extending northward to Tibet [13,14,15]. It is a primate living in both a dark coniferous forest and a mixed coniferous and broad-leaved forest at an altitude of 3000–4000 m [5,13,16]. It has the highest altitude distribution of any non-human primate [5,13,16]. The high-altitude virgin forests to which its habitat belongs are a natural heritage site and contain the richest biodiversity of any temperate region in the world [17,18]. It is globally recognized for its high levels of species diversity, as well as for its wide range of habitats and ecosystems. Amongst the 30 ecosystems identified within Yunnan according to the Chinese classification and the 114 forest types found here, there are large numbers of endemic, threatened, and rare species. Although it only comprises 4.1% of China’s total area, Yunnan contains a disproportionate amount of its biodiversity [19].

In recent years, this narrow area between the Lancang and Jinsha rivers has experienced habitat degradation, fragmentation, and significant loss of biodiversity [5,8,18]. This impact correlates with human population growth and increasing economic development around the habitat area through activities such as farming, logging, and hunting [8,16,19,20,21,22]. More than 10 ethnic minorities, including Tibetan, Bai, Naxi, Yi, Pumi, Lisu, Nu, and Dulong, live in and around the study area. They live in poverty, have poor access to transportation, and realize low productivity from the land. Crop farming and animal grazing are the main production methods for their survival. This introduces significant conflict between community livelihoods and Yunnan snub-nosed monkey conservation [23,24,25].

The primary reasons for the decline in habitat quality include human activities and significant changes in land use patterns [1,12,26,27,28]. The habitat quality module of the InVEST model is effective at quantifying a habitat’s quality by evaluating habitat quality indicators [11,28]. These indicators include land cover types, threat factors, and the rate of decrease in the threat factor impact as the distance from the threat factor increases [11,28,29]. The InVEST model has more flexibility, higher reliability, easier access to data, and more easily visualized results when compared with other models [9,29,30,31]. The InVEST model has been used in habitat quality studies to address a wide range of questions with particular focus on species conservation (e.g., shorebirds in the Yellow River Delta [32], the Dantoo crane in the wetlands of Yancheng, Jiangsu [33], and grassland depression birds in North America [34]).

This paper uses the InVEST model to analyze the dynamic changes in the habitat quality of the Yunnan snub-nosed monkey from 1975 to 2022, aiming to discuss the following issues: (1) how to determine the distance threshold value (the distance at which a land use type threat impact has fallen to 5% of the full value at the source) of the threat factor to improve the accuracy of the model, (2) how the habitat quality, habitat degradation, and habitat rarity in the study area has changed in the study period, and (3) discuss the impact of changing habitat quality on the population of Yunnan snub-nosed monkeys and put forward protection strategies for the whole territory of these monkeys.

By using data on anthropogenic threats, land use, and expert knowledge, the InVEST model can be used to obtain reliable indicators of current biodiversity responses to threats and identify priority areas for conservation [1,12,30,35].

Previous studies on the habitat quality of the Yunnan snub-nosed monkey only carried out preliminary discrimination and analysis based on monkey colony corridors and potential suitable habitats due to the particularity of species distribution and the limitation of data acquisition [5,36,37,38,39,40]. Previous research using the InVEST model to statically model habitat quality studied the impact of villages on monkey habitat quality [12]. This paper improves on that by looking at the habitat quality of the study area over several decades while also expanding the study area to include the contiguous habitat which extends into Tibet, providing a more complete set of data for analysis.

## 2. Materials and Methods

### 2.1. Study Area and Species

The study area is located at the Three Parallel Rivers region (98°37′–99°41′ E, 26°14′––29°20′ N) [8]. Specifically, it includes Deqin, Weixi, Lijiang, Jianchuan, Lanping, and Yunlong Counties in Yunnan, and Kangman County in Tibet, with a total area of 17,026 km^2^.

Monkey populations were obtained from previous field investigation work which occurred in 2007 and 2022 [8,41]. They were surveyed by tagging some monkeys with GPS-based tracking devices and by photographing them [8,41]. Though the number of groups between the two investigations are different (15 groups in 2007 and 17 in 2022), their geographic locations are nearly identical, providing a good basis for comparison analysis [8,41]. The data from these field investigations were mapped against the study area using ArcGIS 10.2 to show the home range of each monkey group (Figure 1).

### 2.2. Land Use and Land Cover (LULC) and Normalized Difference Vegetation Index (NDVI)

LULC data for 1975, 1990, 2000, and 2022 were obtained from a supervised classification of SPOT-5 images with ground-truthing by the Conservation Information Centre of The Nature Conservancy’s China program [12]. Each land cover type was assigned a Habitat Suitability rating for Yunnan snub-nosed monkeys. These ratings are: most suitable (with a value of 1.0), suitable (0.8), less suitable (0.6), unsuitable (0.2), and obstructive (0.0) [12,37].

NDVI data from 1975, 1990, 2000, and 2022 were obtained from four-phase Landsat image data (https://earthexplorer.usgs.gov/ (accessed on 10 December 2022)) calculated using the maximum value synthesis method. Maximum value synthesis is a standard method for improving the accuracy of land coverage classification from satellite images by reducing or eliminating interference from factors such as cloud coverage and air pollution [42].

As both NDVI and LULC data were obtained from image interpretations of the same areas, the LULC and NDVI change in concert with each other [42].

### 2.3. Habitat Quality Assessment

The InVEST model evaluates habitat quality by computing a habitat quality score based on land use, land cover, and biodiversity stress factors across various LULC types in a defined area along with the respective degrees of degradation of these types over time. Habitat rarity score (the relative commonness of the habitat state compared to a baseline state of the same location) is also calculated using the InVEST model. The resulting habitat quality score (sometimes simply referred to as habitat quality) and habitat rarity score (sometimes simply referred to as habitat rarity) can be used together to reflect biodiversity within the analyzed area [9,29,30,31,43].

The habitat quality score is determined using the following equations:$$Q_{\mathrm{xj}}=H_{j}\left(1-\left(\frac{D_{\mathrm{xj}}^{z}}{D_{\mathrm{xj}}^{z}+k^{z}}\right)\right)$$

$Q_{\mathrm{xj}}$ indicates the quality of habitat in parcel x that is in LULC j; $H_{j}$ is the habitat suitability of LULC type j; z is a scaling parameter set at 2.5; k is the half-saturation constant set at 0.5; $D_{\mathrm{xj}}$ indexes the total threat level in grid cell x with LULC type j [37,43].
$$D_{\mathrm{xj}}=\overset{R}{\underset{r=1}{\sum }}\overset{Y_{r}}{\underset{y=1}{\sum }}\left(\frac{w_{r}}{\sum_{r=1}^{R}w_{r}}\right)r_{y}i_{\mathrm{rxy}}\beta_{x}S_{\mathrm{jr}}$$

$D_{\mathrm{xj}}$ indexes the total threat level in grid cell x with LULC type j; y indexes all grid cells on r’s raster map; $Y_{r}$ indicates the set of grid cells on r’s raster map; $w_{r}$ is the degradation source’s weight, indicating the relative destructiveness of a degradation source to all habitats ($w_{r}\in \left[0,1\right]$); $i_{\mathrm{rxy}}$ indexes the impact of threat r that originates in grid cell y on habitat in grid cell x; $\beta_{x}$ is the level of accessibility in grid x, ranging from 0 (no accessibility) to 1 (complete accessibility) and represents the legal, institutional, social, and/or physical protection of the location (low protection = high accessibility); $S_{\mathrm{jr}}\in \left[0,1\right]$ indicates the sensitivity of LULC j to threat r (values closer to 1 indicate greater sensitivity). We used the model defaults for $\beta_{x}$ [37,43].
$$i_{\mathrm{rxy}}=\exp\left(-\left(\frac{2.99}{d_{\mathrm{rmax}}}\right)d_{\mathrm{xy}}\right)$$

$i_{\mathrm{rxy}}$ indexes the impact of threat r that originates in grid cell y on habitat in grid cell x; $d_{\mathrm{xy}}$ is the linear distance between grid cells x and y; $d_{\mathrm{rmax}}$ is the distance threshold value of threat r’s reach across space [43].
$$R_{j}=1-\left(\frac{N_{j}}{N_{\mathrm{jbaseline}}+N_{j}}\right)$$

$R_{j}$ is rarity of LULC type j, compared to a baseline map; $N_{j}$ the area of grid cells of LULC j on the current map; $N_{\mathrm{jbaseline}}$ gives the area of grid cells of LULC j on the baseline map [43].
$$R_{x}=\overset{j}{\underset{x=1}{\sum }}\delta_{\mathrm{xj}}R_{j}$$

$R_{x}$ is the overall rarity of habitat type in grid cell x; $\delta_{\mathrm{xj}}$ = 1 if grid cell contains LULC type j on the current map, and =0 otherwise; $R_{j}$ is rarity of LULC type j, compared to a baseline map [43].

While the InVEST model may appear to have intended its use to be for impacts of edge effects, we believe this model is appropriate for analyzing the habitat for a single species. This is because of the spatial nature of patterns in biodiversity, because LULC maps represent conditions spatially, and because threats to the habitat which are included in the model have a ranging impact, and hence also have spatial impact. The InVEST model applies the spatial threat impacts against the LULC map to provide maps on habitat quality and habitat degradation (or perhaps habitat improvement, if that is the case) [43]. Thus, a single-species habitat quality area can be evaluated using the InVEST model.

We used LULC data from 1975 to 2022, selecting Other Non-Forestry Land, Artificial Economic Forest, Cropland, and Artificial Construction as the threat factors to habitat quality to estimate the 1975 to 2022 habitat quality of the Yunnan snub-nosed monkey distribution area. While there are other threat factors, such as roads and degree of community development, such threat factors tend to overlap [12,20,39]. Hence, we focused on these four as a balance between including all threats and not counting the same threat multiple times [12,20,39].

We standardize the LULC raster data and the threat source raster data, unifying them into a 30 m WGS 84 coordinate system. The weight of threat factors and the suitability and sensitivity parameters of land cover types were determined by combining literature research, discussion from experts in the areas of animal ethology and primates, discussion from workers in protected land areas, and the actual situation of the Yunnan snub-nosed monkey habitat through field surveys. Each land cover type was assigned a Habitat Suitability rating for Yunnan snub-nosed monkeys, with values ranging from 1.0 to 0.0, as seen in Table 1 [37]. The parameter values were then processed to obtain the sensitivity of land cover types to threats [11,38,44,45] (Table 1).

The distance threshold value is based on the maximum correlation coefficient between the NDVI and the land use type for that point in time. The values of NDVI and the area of threat factors (Other Non-Forestry Land, Economic Forest, Cropland, Artificial Construction) in 1975, 1990, 2000, 2020 with buffer zones of 1 km, 1.5 km, 2 km, 2.5 km, 3 km, 3.5 km, 4 km, 4.5 km, and 5 km radius were processed through ArcGIS10.8.

Cluster analysis was conducted on the NDVI values and the threat factors of the area to eliminate redundant variables and abnormal values in the scatter plot [46]. Through correlation analysis, we investigated whether there was any relationship between threat factors (Land Use) and NDVI of surrounding grid squares at various distances away from the grid square and what the strength of the relationship was [47]. If the Pearson correlation coefficient was greater than 0.5 (the larger the correlation, the more correlated it is) and the two-tailed correlation was less than 0.05 (excluding chance and passing the test), the correlation was high [47]. Because it was not necessary to determine the specific mathematical form of correlation between variables, an exponential regression or GLM was not performed [47,48] (Table 2).

Though we did compare the changes in habitat quality over time to monkey population changes over nearly the same time, we did not deeply analyze the relationship between the habitat quality score and monkey population size because previous studies and research findings have shown the significance of habitat changes (especially loss and fragmentation) as causes of population declines in animal species, and particularly so in primates because of their high habitat dependence [49].

## 3. Results

### 3.1. Distance Threshold of Threat Factors

During the study period, the distance threshold of the impact of each threat factor trended upward. This distance threshold has more than tripled between 1975 and 2022 for every threat factor (Table 2).

### 3.2. Temporal and Spatial Changes in Habitat Degradation in the Distribution Area of Yunnan Snub-Nosed Monkey

The degree of habitat degradation in the distribution area of the Yunnan snub-nosed monkey has increased over the study period, with the key areas of habitat degradation primarily located in the villages in the more southern and more northern portions of the study area, though in different ways. The more southern portions have experienced degradation across the widest area, whereas the more northern portions have experienced the most severe degradation (Figure 2).

The spatial statistics of the degrees of habitat degradation show an overall upward trend. Further, the maximum value of the habitat degradation degree rose from 1.2160 to 1.2708. The primary degradation occurred between 1975 and 1990. There was a small decrease (improvement) between 1990 and 2000 in both the average degradation and highest degradation; however, between 2000 and 2022 the degradation increased again, exceeding 1990′s numbers (Table 3).

The areas of Highest, High, and Medium Degradation have increased significantly. The area of High Degradation shows the largest growth rate, from 1888.42 km^2^ in 1975 (11.09% of the area) to 4951.21 km^2^ (29.08%) in 2022. The areas of Low and Lowest Degradation decreased, with the area of Low Degradation decreasing the most, down from 36.34% in 1975 to 12.92% in 2022 (Table 4).

### 3.3. Spatial and Temporal Variation of Habitat Quality in Yunnan Snub-Nosed Monkey

During the study period, the overall spatial distribution characteristics show the area of low habitat quality is increasing across the board, though the northern sections have consistently significantly higher habitat quality than the southern sections (Figure 3). The striking decline in habitat quality seen in the northern peninsulas of land between 1975 and 1990 was primarily due to the aggressive increase in economic forestry, cropland, artificial construction, and other non-forestry land uses by the local population. We determined this by comparing the land use and habitat maps of 1975 and 1990 with GIS spatial superposition and observed a significant increase in these land use types in these areas.

The Habitat Quality Score average trended downward over the study period, indicating a decline in the overall Habitat Quality for the area. The average habitat quality dropped from 0.4880 to 0.4223 between 1975 and 2022, with the period from 1975 to 1990 seeing the biggest drop (~82% of the total drop). There was a slight improvement between 1990 and 2000. However, that was short lived, with 2022 showing an average lower than 1990. The standard deviation of the Habitat Quality Score also dropped, reducing from 0.3626 in 1975 down to 0.3382 in 2022, indicating a tightening of the range of Habitat Quality over time. Unlike the average quality, the standard deviation continued to decline over each of the years (Table 5).

The areas of highest and high habitat quality are decreasing over time. The area of high habitat quality has the largest decline rate, from 2784.80 km^2^ (16.36% of the area) in 1975 to 2041.21 km^2^ (11.99%) in 2022. The moderate habitat quality area remains consistent (±0.68%) over the study period. The areas of low and lowest habitat quality show an increasing trend, with the area of low habitat quality having the largest growth, rising significantly from 2.76% in 1975 to 7.96% in 2022. For both lowest- and low-quality areas and highest- and high-quality areas, most of the change occurred between 1975 and 1990. Low-quality areas went up and down between 1990 and 2022, but overall did not change much. The highest quality areas went up slightly between 1990 and 2000 before dropping by nearly 1% between 2000 and 2022. The lowest- and high-quality areas increased and decreased, respectively, throughout the entire time (Table 6).

### 3.4. Temporal and Spatial Variations in Habitat Rarity

Habitat rarity across the study area has increased (degraded) over the study period of 1975 to 2022. The 1975–1990 and 2000–2022 habitat rarity increases are happening primarily in the southern region. However, we also see significant rarity increases in both 1975–1990 and 1990–2000 in a central “spine” portion of the northern area (Figure 4). This increase is primarily due to the local population’s increase in other non-forestry land use in that area. We determined this by comparing the land use and habitat maps 1975, 1990, and 2000 with GIS spatial superposition and observed a significant increase in this land use type in that area.

From 1975 to 1990 we see a rarity increase (including significant rarity) in 46.12% of the area, with significant rarity alone showing an increase in 4.30% of the area. Additionally, 51.01% of the area decreased or significantly decreased in rarity. From 1990 to 2000, habitat rarity increased in nearly the entire study area (95.16%, including significant rarity). From 2000 to 2022, we see improvement via a drop back down to 49.62% of the study area increase in habitat rarity and no areas of significant increase in rarity. However, there also were no areas of significant decrease in rarity (Table 7). The ranges of Habitat Rarity Scores used to denote each category were informed by the Habitat Rarity section of the Habitat Quality model information in the InVEST model user’s guide [43].

### 3.5. Dynamic Analysis of Habitat Quality

The habitat quality of most of the monkey groups (11 out of 18) improved between 2000 and 2022. The average value of the Habitat Quality Score rose from 0.7384 to 0.8175. The improvement of the habitat quality of the 11 monkey groups that had improved habitats will be beneficial to the reproduction of the Yunnan snub-nosed monkey. However, the habitat quality of seven monkey groups (Xiaochangdu, Milaka, Gehuaqing, Shikuadi, Anyi, Baijixun, and Longma) is declining. Additionally, while the habitat quality for the Bamei group did increase, this increase was nominal. These eight monkey groups should receive more attention in future research. The Dapingzi group’s habitat quality improvement was also nominal, but as this group was extinct by 2022, it cannot be included in future research (Table 8 and Table 9).

## 4. Discussion

The degree of habitat degradation reflects the degree of impact of habitat threat factors on land cover types. The higher the degree of habitat degradation, the greater the possibility of habitat destruction in the future.

We speculate that the habitat degradation of the Yunnan snub-nosed monkey from 1975 to 2022 may largely be due to recent societal and economic developments of the human populations in and around the study area, which consist of many ethnic minorities who traditionally relied on hunting for their survival [23,24,25]. This societal and economic shift has led to an increase in human developments and expansion of land use types that are a threat to the monkey’s habitat, increasing habitat fragmentation and reducing habitat quality [12]. Historically, the monkey population’s key threat factor was hunting by humans [23,24,25]. Today, habitat loss and fragmentation are the monkey population’s key threat factors [15,22].

To help protect and improve these habitats for the future, monitoring of habitats needs to be strengthened and the habitats of Yunnan snub-nosed monkeys in the whole territory need to be integrated with sympatric species. Providing habitat corridors to connect isolated habitat areas is one approach that could help.

In the InVEST model, habitat quality and habitat rarity act as proxies for biodiversity, ultimately estimating the extent of habitat and vegetation types across a landscape, and their state of degradation. While mapping habitat quality can help to identify areas where biodiversity is likely to be most intact or imperiled, it is also critical to evaluate the relative rarity of habitats on the landscape regardless of quality. In many conservation plans, habitats that are rarer are given higher priority because options and opportunities for conserving them are limited and if all such habitats are lost, so too are the species and processes associated with them. Through the superimposed analysis of the distribution range, the LULC category and habitat scarcity map, the land use planning can be further protected by Yunnan snub-nosed monkeys [43,49].

Habitat quality scores reflect the degree of fragmentation of habitat patches and the ability of habitats to resist interference from threats. The habitat degradation scores are based on the land use and land cover (LULC) surrounding each grid square, not the quality of the grid square itself (NDVI). The InVEST Habitat Quality model combines information on LULC and threats to biodiversity to produce habitat quality maps, ultimately estimating the extent of habitat and LULC types across a landscape and their state of degradation. Studies have shown that because the southern region is dominated by cropland and artificial construction, the habitat quality is relatively poor. We have seen in this paper that this area has continued to decline in quality over the entire study period. With the acceleration of urbanization, the degradation of habitat quality is also accelerating. Future protection should give priority to such low habitat quality areas.

Additionally, priority should be given to key monkey groups which are experiencing habitat quality reduction (Xiaochangdu, Milaka, Longma, and Anyi) and geographic isolation (Bamei). Further, the Gehuaqing and Shikuadi monkey groups should be prioritized, as they are new and potentially at higher risk of population instability.

Habitat rarity reflects the degree of change in fragmentation of land coverage types and the stability of regional ecological security pattern [43,50]. The higher the habitat rarity score, the larger the degree of change. If habitat patches of the land coverage type are damaged, the habitat quality and the stability of the ecological security pattern in the region may be severely adversely affected. Rarer habitats require higher conservation priority because the options and opportunities to protect them are very limited, and if all such habitats disappear, the species and ecosystem processes associated with them will also disappear.

Steps to take to improve habitat quality in the study area and promote genetic diversity across monkey groups include:Increasing the connectivity of habitat patches through the building and fostering of habitat corridors, especially improving connectivity of lower-quality habitat areas to those of higher habitat quality.Reversing the severe fragmentation of the monkey groups of low-quality habitat patches by improving the habitat quality between the low-quality patches beyond building habitat corridors.Reducing the threat impact (both distance and degree) of existing threats from human activities in and around the study area, such as current crop production and existing buildings.Eliminating the threat impact of new land use when it is desired by the human population, such as if people are looking to increase crop production or build new structures. This needs to be a key part of the initial planning process, as well as an ongoing part of the land use execution.Beginning or increasing community participation, friendly development, and public involvement to work toward a sustainable and healthy increase in snub-nosed monkey habitat quality and an overall improvement in habitat ecosystem health [14].Promoting community and public participation in the conservation of snub-nosed monkey habitat quality. Exploring Yunnan snub-nosed monkey-friendly community development, carrying out Yunnan snub-nosed monkey-friendly community development projects, creating a brand with the theme of Yunnan snub-nosed monkey habitat conservation, developing ecological agricultural products and traditional handicrafts, and assisting with connecting to external markets [51].Improve education through constructing a nature center for Yunnan snub-nosed monkeys, conducting nature education and ecological guided tours on the theme of Yunnan snub-nosed monkeys to benefit from public participation in conservation, creating more public awareness of Yunnan snub-nosed monkeys and the importance of their conservation, and attracting potential support to establish a sustainable conservation mechanism [52].

In terms of future research directions, we are working on predicting and planning the quality of snub-nosed monkey habitat under different future land use scenarios and analyzing how climate change and population growth will affect snub-nosed monkey populations, ecosystem services, and biodiversity. We are also working on helping to provide strategies to address the future conservation of the species to protect the diversity of biological activities, improve human livelihoods, delineate the ideal range for promoting harmonious coexistence between humans and nature, and design permissible mitigation options for economic development and species conservation.

The use of InVEST not only allows for more accurate determination and visualization of the range of each class of habitat quality of Yunnan snub-nosed monkeys, but also allows for the conservation of other endangered species. For Sichuan snub-nosed monkeys in Sichuan, the habitat suitability pattern of Sichuan snub-nosed monkeys was constructed by calculating each variable factor affecting the habitat through the maximum entropy model [53]. For giant pandas in China, habitat models were established through field surveys as well as remote sensing data, and several different scenarios were simulated to delineate protection red lines covering different proportions of giant panda habitat, which can both effectively improve habitat connectivity and constrain human interference [54]. For elephants in India, the geospatial components of the landscape were fully considered to delineate suitable and unsuitable habitats for elephants, helping to avoid human conflicts and protecting the elephants [55]. In some areas where habitats were damaged due to human interference and industrialization, connectivity models were used to maximize the connection to the number of habitats [56]. As seen through these conservation measures, the use of InVEST can not only more accurately determine the habitat quality of each preferred land use type of Yunnan snub-nosed monkeys and determine suitable habitat, but also visualize it, making it easier to delineate areas more accurately for protection.

## 5. Conclusions

The changes in habitat degradation, habitat quality, and habitat rarity of the Yunnan snub-nosed monkey were analyzed from 1975 to 2022. To accomplish this, the Land Use and Land Cover data from satellite images, as well as NDVI data from satellite images, were extracted for the study area from four points in time—1975, 1990, 2000, and 2022. We derived the habitat suitability and threat sensitivity of the Land Cover types. We determined the maximum distances that threats affected land cover types, by how much, and in what mathematical manner they decrease over distances (linear or exponential). These data were processed through the InVEST model, which finally provided us with the habitat degradation, habitat quality, and habitat rarity data.

By conducting this analysis for multiple points in time, we were able to look at trends over time across the study area and compare this information to the change in monkey populations throughout the study area between 2007 and 2022. This change in monkey populations included the extinction of one of the monkey groups and the emergence of two new groups.

From these computations and analyses, we identified areas for focus and recommended actions to take to help slow, stop, and hopefully, in time, reverse the ongoing habitat degradation which continues to put the Yunnan snub-nosed monkey populations at risk.

In addition to the specific findings in this paper, we explored a new method of evaluating the habitat quality of species via the InVEST model, which lays an important foundation for accurate and scientific evaluation of habitat quality and can be an example for the research on other species.

## Figures and Tables

**Figure 1 biology-12-00886-f001:**
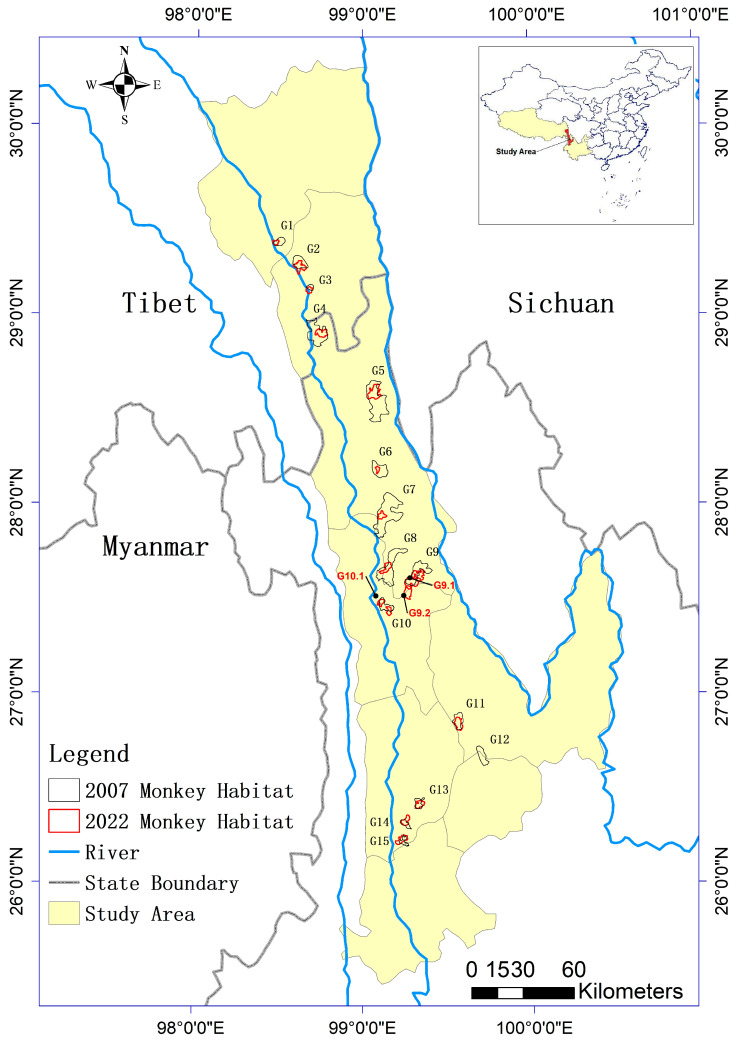
Home ranges of the monkey groups from 2007 and 2022 within the study area.

**Figure 2 biology-12-00886-f002:**
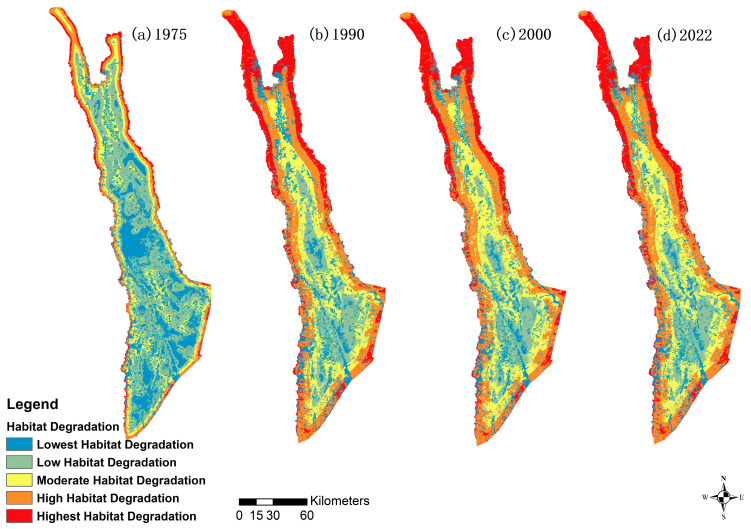
Mapping and trend of habitat degradation.

**Figure 3 biology-12-00886-f003:**
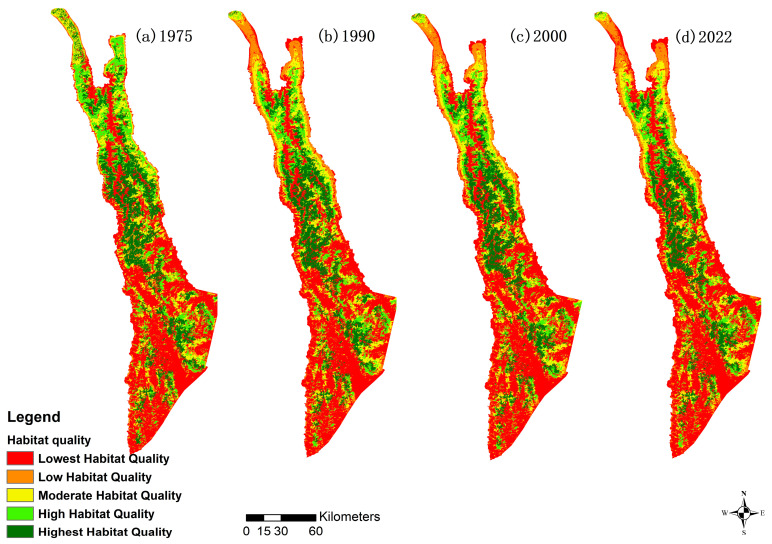
Mapping and trend of habitat quality.

**Figure 4 biology-12-00886-f004:**
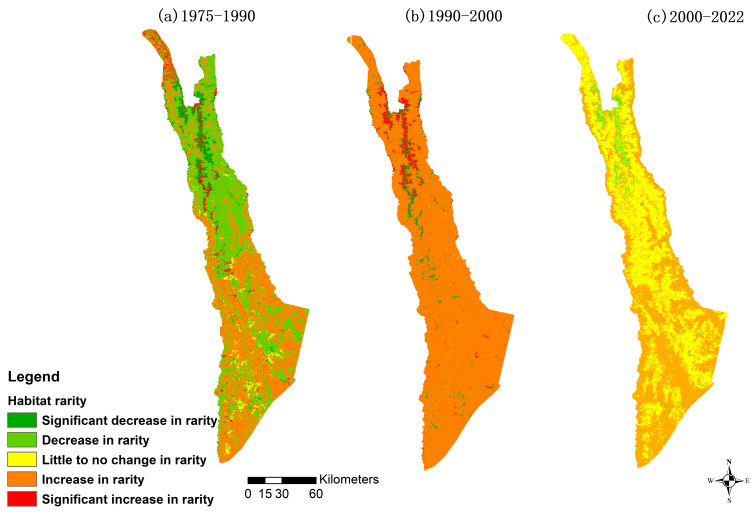
Mapping of changes in habitat rarity.

**Table 1 biology-12-00886-t001:** Sensitivity of land cover types to threats.

	Habitat	Sensitivity to Each Threat			
Land Cover Types	Suitability	ONFL	AEF	Crop	AC
Alpine coniferous forest	0.6	0.4	0.3	0.4	0.8
Shrub	0.8	0.5	0.3	0.4	0.8
Huashan pine	1	0.5	0.4	0.5	0.9
Barren	0.2	0.2	0.1	0.1	0.2
Snow	0.2	0.2	0.1	0.1	0.2
Broadleaf forest	0.6	0.4	0.3	0.4	0.8
Water	0.2	0.2	0.1	0.1	0.2
Hard-leaved evergreen spruce forest	0.8	0.5	0.3	0.4	0.8
Spruce forest	1	0.5	0.4	0.5	0.9
Yunnan pine forest	0.2	0.2	0.3	0.1	0.2
Mixed coniferous forest	1	0.5	0.4	0.5	0.9

Note: ONFL refers to Other Non-Forestry Land, AEF refers to Artificial Economic Forest, Crop refers to Cropland, and AC refers to Artificial Construction.

**Table 2 biology-12-00886-t002:** Threshold distances of threat factors at each point in time.

Threat Factors	Distance Threshold Value (km)			
	1975	1990	2000	2022
Other Non-Forestry Land	1.5	4.5	5	5
Economic Forest	1	3	2	3.5
Cropland	1	3	2	3.5
Artificial Construction	1.5	4.5	5	5

**Table 3 biology-12-00886-t003:** Degree of habitat degradation at each point in time.

Degradation Values	1975	1990	2000	2022
Lowest	0.0000	0.0000	0.0000	0.0000
Average	0.0494	0.1380	0.1353	0.1524
Highest	1.2160	1.2474	1.2338	1.2708
Standard deviation	0.1192	0.1942	0.1867	0.2037

**Table 4 biology-12-00886-t004:** Trends of habitat degradation status.

Habitat Degradation	Degradation Value Range	1975		1990		2000		2022	
		Area (km^2^)	%	Area (km^2^)	%	Area (km^2^)	%	Area (km^2^)	%
Lowest Degradation	<0.003	5154.04	30.27	3007.51	17.66	2853.97	16.76	2804.05	16.47
Low Degradation	≥0.003, <0.016	6186.69	36.34	2726.33	16.01	2280.16	13.39	2199.17	12.92
Moderate Degradation	≥0.016, <0.069	3000.05	17.62	3876.97	22.77	4292.63	25.21	3987.17	23.42
High Degradation	≥0.069, <0.285	1888.42	11.09	4672.36	27.44	4975.88	29.22	4951.21	29.08
Highest Degradation	≥0.285	797.06	4.68	2743.10	16.11	2623.63	15.41	3084.66	18.12

**Table 5 biology-12-00886-t005:** Habitat Quality Score at each point in time.

Habitat Quality Score	1975	1990	2000	2022
Average	0.4880	0.4348	0.4359	0.4233
Standard deviation	0.3626	0.3442	0.3423	0.3382

**Table 6 biology-12-00886-t006:** Trends of habitat quality status.

Habitat Quality Status	Habitat Quality Score Range	1975		1990		2000		2022	
		Area (km^2^)	%	Area (km^2^)	%	Area (km^2^)	%	Area (km^2^)	%
Lowest	<0.2	7714.32	45.31	7972.65	46.83	8025.75	47.14	8137.49	47.79
Low	≥0.2, <0.4	469.54	2.76	1232.95	7.24	1151.79	6.76	1355.57	7.96
Moderate	≥0.4, <0.6	2562.49	15.05	2640.68	15.51	2679.01	15.73	2626.50	15.43
High	≥0.6, <0.8	2784.80	16.36	2184.11	12.83	2157.00	12.67	2041.21	11.99
Highest	≥0.8	3495.11	20.53	2995.89	17.60	3012.71	17.69	2865.50	16.83

**Table 7 biology-12-00886-t007:** Trends of habitat rarity status.

Habitat Rarity Status	Habitat Rarity Score Range	1975–1990		1990–2000		2000–2022	
		Area (km^2^)	%	Area (km^2^)	%	Area (km^2^)	%
Significant decrease	<−0.100	684.53	4.02	783.56	4.60	0.00	0.00
Decrease	≥−0.100, <−0.001	8000.68	46.99	0.00	0.00	628.59	3.69
Little to no change	≥−0.001, <+0.001	488.29	2.87	39.28	0.23	7949.53	46.69
Increase	≥+0.001, <+0.100	7120.37	41.82	15,570.92	91.45	8448.16	49.62
Significant increase	≥+0.100	732.40	4.30	632.51	3.71	0.00	0.00

**Table 8 biology-12-00886-t008:** Changes in habitat quality of bands of *R. bieti* between 2000 to 2022.

ID	Site	Population of *R. bieti*		Habitat Quality Score	
		2007	2022	2000	2022
G1	Zhina	50	80	0.6295	0.6521
G2	Xiaochangdu	>200	280	0.3242	0.2389
G3	Milaka	100	60	0.2771	0.2261
G4	Bamei	80	100	0.5345	0.5452
G5	Wuyapuya	>300	400	0.6911	0.8022
G6	Cikatong	50	50	0.7911	0.8677
G7	Guyoulong (guilong)	100	80	0.7723	0.9642
G8	Shiba	200	200	0.8156	0.9577
G9	Guomorong (Xiangguqing)	>900	480	0.7863	0.8661
G9.1	Gehuaqing	——	450	0.7184	0.7064
G9.2	Shikuadi	——	120	0.7074	0.6954
G10	Akou (Anyi)	30	40	0.8693	0.8403
G10.1	Baijixun (Yongan)	——	40	0.7411	0.6934
G11	Jinsichang	250	310	0.8797	0.9399
G12	Dapingzi	<50	0	0.7782	0.7808
G13	Longma	120	140	0.5377	0.5099
G14	Lashashan	100	130	0.6450	0.7804
G15	Neidaqin (Fuhe)	>100	120	0.7989	0.8367

**Table 9 biology-12-00886-t009:** Impact of the order of magnitude of change in the Habitat Quality Score.

Change in Score	Impact
−0.9999 to −0.1000	Significant degradation—urgent action needed
−0.0999 to −0.0100	Impactful degradation—needs addressing soon
−0.0099 to −0.0010	Weak degradation
−0.0009 to +0.0009	Negligible impact
+0.0010 to +0.0099	Weak improvement
+0.0100 to +0.0999	Impactful improvement—things are slowly getting better
+0.1000 to +0.9999	Significant improvement—things are rapidly getting better

## Data Availability

Data are available on request.

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
