# Peer review of "The Change in Habitat Quality for the Yunnan Snub-Nosed Monkey from 1975 to 2022"

_biology, 2023, doi:10.3390/biology12060886_

Round 1
Reviewer 1 Report
The manuscript focused an interesting issue and reported a dynamic study in support of a critically endangered primate, the Yunnan snub-nosed monkey. The manuscript used the InVEST model to analyze the dynamic changes in the habitat quality of the Yunnan snub-nosed monkey from 1975 to 2022. Based on species habitat suitability, it introduced a new perspective for impacts on the habitat for the conservation of other species. However, there was still some improvement in the quality of the paper before it is accepted.
Point 1: L45-55, In Introduction, the authors should add the information on how the ethnic minorities affects the habitat of Yunnan snub-nosed monkeys.
Point 2: L89-103, In Study area and species, the authors should add the information about how to survey on the population of Yunnan snub-nosed monkeys.
Point 3: L120, How to calculate the Habitat Quality score ?
Point 4: L159, How to calculate the ‘Sensitivity to each threat’ score ?
Point 5: L159, It is recommended not to use abbreviations (‘Qtfld’, ‘Rgjjl’, ‘Nmd’, and ‘Rgjz’), which are not conducive to reading and understanding.
Author Response
Point 1: L45-55, In Introduction, the authors should add the information on how the ethnic minorities affects the habitat of Yunnan snub-nosed monkeys.
Response 1: Changes have been made to section 1. Introduction to address this.
Point 2: L89-103, In Study area and species, the authors should add the information about how to survey on the population of Yunnan snub-nosed monkeys.
Response 2: Changes have been made to section 2.1. Study area and species to address this.
Point 3: L120, How to calculate the Habitat Quality score ?
Response 3: Changes have been made to section 2.3. Habitat Quality Assessment to address this.
Point 4: L159, How to calculate the ‘Sensitivity to each threat’ score ?
Response 4: Changes have been made to section 2.3. Habitat Quality Assessment to address this.
Point 5: L159, It is recommended not to use abbreviations (‘Qtfld’, ‘Rgjjl’, ‘Nmd’, and ‘Rgjz’), which are not conducive to reading and understanding
Response 5: Changes have been made to Table 2 to address this.
Reviewer 2 Report
Overall, this is an interesting paper that provides new information about habitat degradation in areas occupied by the Yunnan snub-nosed monkey, a declining species of clear conservation concern. The paper is of general interest, as is the authors’ approach to their data
The relationship between change in habitat quality score and monkey population size should be analyzed, unless there is a compelling reason why this is not possible. This is the primary focus of the paper. Also, the scope of the introduction and discussion should be broadened, as described below.
Specific comments by section
Simple abstract:
A bit unclear. Do you mean ‘find opportunities to address conservation concerns while minimizing potential costs to human economic development? Also, how are you distinguishing between ‘population changes’ and ‘rarity changes?’ And the changes you observed actually increased habitat rarity for the monkey, so ‘adversely affected’ is confusing here; I understand what you mean, but it is better to be direct for clarity.
Abstract:
The phrase ‘increase in habitat rarity’ is not clear here. Do you mean rarity of high-quality habitat, or habitat where the monkeys can potentially survive?
Introduction:
To broaden the scope of this paper, start with a brief discussion of the importance of habitat changes as causes of population declines in animal species, or in primates in general if you choose to restrict it to that taxon. The Introduction is too narrowly focused on a single species but could easily be made broadly relevant.
Please provide the area where this monkey is found earlier in this section, i.e. its current – and ideally also its former – range, so that readers will know where you are writing about. The map you provide is excellent but does not show the species range. How much of its range is in the focal area?
L. 44. In exactly what sense does this area have the highest biodiversity? Be specific here, and perhaps cite some of the commonly most cited papers about the exceptional biodiversity (of plants) in the mountains of China.
Ll. 65-71. The detailed, long list of studies that have used InVEST is not necessary – just cite the papers as examples. E.g., InVEST has been used in varied habitats to address a wide range of questions, with a particular focus on assessment of habitat quality for species of conservation concern (e.g. grassland birds in North America (33), etc.).
The paragraphs that begin on L. 73 and L. 79 are confusing, in part because you start discussing this paper at the end of that second paragraph but then introduce this paper again in the subsequent paragraph. It would be clearer if you first stated what you are presenting here, i.e. the material that is mostly given in the paragraph starting on l. 85, and then explain why it will provide new information. Then your introduction to your own paper will be clear, as will the need for this research.
L. 87. Define ‘distance threshold value’ here, to improve clarity and accessibility for your article.
Materials and Methods
L. 97. Confusing. Do you mean the locations of populations were obtained from previous work? Were the authors of this paper the ones who did that earlier research? What is ‘field investigation reporting?’
Section 2.3: It is difficult to write these kinds of model explanations clearly, but do not capitalize words in the middle of sentences, e.g. ‘Where’ in l. 139 and l. 142. Also, it’s better to use a capital letter at the start of each sentence, e.g. rather than ‘z is a scaling parameter…,’ use this construction: The variable z is a scaling parameter. This will make it much easier for someone unfamiliar with your model to follow your explanation.
L. 139(-140) is a sentence fragment. Or maybe it’s meant to complete the above equations? If so, then the capital letter at the beginning is confusing.
L. 152. ‘tend to overlap with the variables we selected (cite reference for this, if this is what you mean; explain what you do mean here if my interpretation is not correct)
L. 155. ‘The weight of threat factors and the suitability and sensitivity parameters of land cover types were scored by experts ‘ What experts? How did they justify the weights for the different threat factors? This is a critical step and should be better explained in a few sentences.
L. 167-169. Re-write for clarity. What are ‘the statistics corresponding to the values…’ here?
Fig 2: Excellent and highly informative maps. Revise legend for clarity.
Results
Ll. 191-192. Both the maps and the numbers in Table 5 show that the primary degradation occurred between 1975 and 1990. This should be mentioned in Results. Also, there is not a consistently ‘increasing rate of habitat degradation,’ given that by these values both average degradation and highest degradation appear to have decreased between 1990 and 2000.
Similarly, values for areas at different levels of habitat degradation reveal complex patterns that should be described rather than just lumped together as depicting increasing areas of greater degradation. Again, the area suffering from the highest degradation increased dramatically between 1975 and 1990, then remained relatively stable in 2000, and concomitantly, the area with the lowest level degradation decreased greatly between 1975 and 1990, decreased at a lower rate between 1990 and 2000, and then was stable through 2022. The authors note this later in the manuscript in their ‘Dynamic Analysis of Habitat Quality,’ which begins on l. 244.
Table 7 is not informative, given that in all years the lowest and highest quality habitats scored at 0 and 1, respectively, meaning only that there were sites with the lowest and the highest quality in each of the years sampled. That can go in the text. Also, if possible with your data, do a statistical test to see whether the changes in standard deviation are significant.
The assessment of trends in habitat quality (Table 8 and ll. 215-220) are useful. However, again nuances in the data should be briefly mentioned: lowest quality habitat increased slowly, low-quality habitat increased sharply between 1975 and 1990 and then was relatively stable, and high- and highest-quality habitats decreased sharply between 1975 and 1990 and then were relatively stable.
Figure 3: Visually, the most striking decline in habitat quality is in the north, where the two ‘peninsulas’ of land turn from greens to yellow/orange/read. Do you know what happened there? This does not entirely agree with your points about habitat rarity in the next section, which is interesting.
Fig. 4: This is visually showing what I have been writing about in the above sections – this figure is very helpful, but it would enrich this paper to describe the earlier figures in more detail as you present them.
Table 10. I am unclear about whether significant differences in habitat quality scores that are measured in hundredths or tenths of points are ecologically meaningful. Could you explain the scale more fully in methods to justify discussion of small differences? Give examples from other studies with similar differences in habitat quality scores?
Discussion
This section is thorough and well conceived. It is very narrowly focused on the analyses and findings of this paper, however, and the paper would be stronger and of greater general interest if the findings here were compared to related work on other species and/or habitats of conservation concern.
Author Response
Point 1: The relationship between change in habitat quality score and monkey population size should be analyzed, unless there is a compelling reason why this is not possible. This is the primary focus of the paper. Also, the scope of the introduction and discussion should be broadened, as described below.
Response 1: Changes have been made to sections 1. Introduction, 2.3. Habitat Quality Assessment, and 4. Discussion to address this.
Point 2: Simple abstract: A bit unclear. Do you mean ‘find opportunities to address conservation concerns while minimizing potential costs to human economic development? Also, how are you distinguishing between ‘population changes’ and ‘rarity changes?’ And the changes you observed actually increased habitat rarity for the monkey, so ‘adversely affected’ is confusing here; I understand what you mean, but it is better to be direct for clarity.
Response 2: Changes have been made to the Simply Summary section to address this.
Point 3: Abstract: The phrase ‘increase in habitat rarity’ is not clear here. Do you mean rarity of high-quality habitat, or habitat where the monkeys can potentially survive?
Response 3: Changes have been made to the Abstract section to address this.
Point 4: Introduction: To broaden the scope of this paper, start with a brief discussion of the importance of habitat changes as causes of population declines in animal species, or in primates in general if you choose to restrict it to that taxon. The Introduction is too narrowly focused on a single species but could easily be made broadly relevant.
Response 4: Changes have been made to section 1. Introduction to address this.
Point 5: Please provide the area where this monkey is found earlier in this section, i.e. its current – and ideally also its former – range, so that readers will know where you are writing about. The map you provide is excellent but does not show the species range. How much of its range is in the focal area?
Response 5: Changes have been made to section 1. Introduction to address this.
Point 6: L. 44. In exactly what sense does this area have the highest biodiversity? Be specific here, and perhaps cite some of the commonly most cited papers about the exceptional biodiversity (of plants) in the mountains of China.
Response 6: Changes have been made to section 1. Introduction to address this.
Point 7: Ll. 65-71. The detailed, long list of studies that have used InVEST is not necessary – just cite the papers as examples. E.g., InVEST has been used in varied habitats to address a wide range of questions, with a particular focus on assessment of habitat quality for species of conservation concern (e.g. grassland birds in North America (33), etc.).
Response 7: Changes have been made to section 1. Introduction to address this.
Point 8: The paragraphs that begin on L. 73 and L. 79 are confusing, in part because you start discussing this paper at the end of that second paragraph but then introduce this paper again in the subsequent paragraph. It would be clearer if you first stated what you are presenting here, i.e. the material that is mostly given in the paragraph starting on l. 85, and then explain why it will provide new information. Then your introduction to your own paper will be clear, as will the need for this research.
Response 8: Changes have been made to section 1. Introduction to address this.
Point 9: L. 87. Define ‘distance threshold value’ here, to improve clarity and accessibility for your article.
Response 9: Changes have been made to section 1. Introduction to address this.
Point 10: Materials and Methods L. 97. Confusing. Do you mean the locations of populations were obtained from previous work? Were the authors of this paper the ones who did that earlier research? What is ‘field investigation reporting?’
Response 10: Changes have been made to section 2.1. Study area and species to address this.
Point 11: Section 2.3: It is difficult to write these kinds of model explanations clearly, but do not capitalize words in the middle of sentences, e.g. ‘Where’ in l. 139 and l. 142. Also, it’s better to use a capital letter at the start of each sentence, e.g. rather than ‘z is a scaling parameter…,’ use this construction: The variable z is a scaling parameter. This will make it much easier for someone unfamiliar with your model to follow your explanation.
Response 11: Changes have been made to section 2.3. Habitat Quality Assessment to address this.
Point 12: L. 139(-140) is a sentence fragment. Or maybe it’s meant to complete the above equations? If so, then the capital letter at the beginning is confusing.
Response 12: Changes have been made to section 2.3. Habitat Quality Assessment to address this.
Point 13: L. 152. ‘tend to overlap with the variables we selected (cite reference for this, if this is what you mean; explain what you do mean here if my interpretation is not correct)
Response 13: Changes have been made to section 2.3. Habitat Quality Assessment to address this.
Point 14: L. 155. ‘The weight of threat factors and the suitability and sensitivity parameters of land cover types were scored by experts ‘ What experts? How did they justify the weights for the different threat factors? This is a critical step and should be better explained in a few sentences.
Response 14: Changes have been made to section 2.3. Habitat Quality Assessment to address this.
Point 15: L. 167-169. Re-write for clarity. What are ‘the statistics corresponding to the values…’ here?
Response 15: Changes have been made to section 2.3. Habitat Quality Assessment to address this.
Point 16: Fig 2: Excellent and highly informative maps. Revise legend for clarity.
Response 16: Changes have been made to Figures 2, 3, and 4 to address this.
Point 17: Results Ll. 191-192. Both the maps and the numbers in Table 5 show that the primary degradation occurred between 1975 and 1990. This should be mentioned in Results. Also, there is not a consistently ‘increasing rate of habitat degradation,’ given that by these values both average degradation and highest degradation appear to have decreased between 1990 and 2000.
Response 17: Changes have been made to section 3.2. Temporal and spatial changes of habitat degradation in the distribution area of Yunnan snub-nosed monkey to address this.
Point 18: Similarly, values for areas at different levels of habitat degradation reveal complex patterns that should be described rather than just lumped together as depicting increasing areas of greater degradation. Again, the area suffering from the highest degradation increased dramatically between 1975 and 1990, then remained relatively stable in 2000, and concomitantly, the area with the lowest level degradation decreased greatly between 1975 and 1990, decreased at a lower rate between 1990 and 2000, and then was stable through 2022. The authors note this later in the manuscript in their ‘Dynamic Analysis of Habitat Quality,’ which begins on l. 244.
Response 18: Changes have been made to section 3.2. Temporal and spatial changes of habitat degradation in the distribution area of Yunnan snub-nosed monkey to address this.
Point 19: Table 7 is not informative, given that in all years the lowest and highest quality habitats scored at 0 and 1, respectively, meaning only that there were sites with the lowest and the highest quality in each of the years sampled. That can go in the text. Also, if possible with your data, do a statistical test to see whether the changes in standard deviation are significant.
Response 19: Changes have been made to section 3.3. Spatial and temporal variation of habitat quality in Yunnan snub-nosed monkey to address this.
Point 20: The assessment of trends in habitat quality (Table 8 and ll. 215-220) are useful. However, again nuances in the data should be briefly mentioned: lowest quality habitat increased slowly, low-quality habitat increased sharply between 1975 and 1990 and then was relatively stable, and high- and highest-quality habitats decreased sharply between 1975 and 1990 and then were relatively stable.
Response 20: Changes have been made to section 3.3. Spatial and temporal variation of habitat quality in Yunnan snub-nosed monkey to address this.
Point 21: Figure 3: Visually, the most striking decline in habitat quality is in the north, where the two ‘peninsulas’ of land turn from greens to yellow/orange/read. Do you know what happened there? This does not entirely agree with your points about habitat rarity in the next section, which is interesting.
Response 21: Changes have been made to section 3.3. Spatial and temporal variation of habitat quality in Yunnan snub-nosed monkey to address this.
Point 22: Fig. 4: This is visually showing what I have been writing about in the above sections – this figure is very helpful, but it would enrich this paper to describe the earlier figures in more detail as you present them.
Response 22: Changes have been made to section 3.4. Temporal and spatial variations in habitat rarity to address this.
Point 23: Table 10. I am unclear about whether significant differences in habitat quality scores that are measured in hundredths or tenths of points are ecologically meaningful. Could you explain the scale more fully in methods to justify discussion of small differences? Give examples from other studies with similar differences in habitat quality scores?
Response 23: Changes have been made to section 3.5. Dynamic Analysis of Habitat Quality to address this.
Point 24: Discussion This section is thorough and well conceived. It is very narrowly focused on the analyses and findings of this paper, however, and the paper would be stronger and of greater general interest if the findings here were compared to related work on other species and/or habitats of conservation concern.
Response 24: Changes have been made to section 4. Discussion to address this.
Reviewer 3 Report
This manuscript takes a new approach to modeling the conservation status of a charismatic endangered primate, adopting a promising new approach using the InVEST habitat quality model to analyze habitat availability and change. InVEST allows the modeler to take account of the impacts of surrounding land use and land cover (LULC) on the habitat suitability of each grid square in the area considered, thus allowing for a more realistic model of the impact of habitat fragmentation. This promises to be a considerable advance in planning for conservation of the Yunnan Snub-nosed Monkey and other threatened species.
1. With the exception of a generic description of the InVEST approach, which is largely paraphrased from the website (below), the Methods section provides sparse detail about what was actually done. For example, the key step of choosing parameter values for habitat suitability is described with a single general sentence on Lines 155-156, without reference: “The weight of threat factors and the suitability and sensitivity parameters of land cover types were scored by experts.” The information from this scoring, which is valuable in itself, is presented in Table 2. However, the description leaves too many questions unanswered: Who were these experts? What were the principles behind the scoring? What information and what instructions were they given?
2. Habitat Degradation: Habitat Degradation here is measured with the index generated by the InVEST model. This rests on the LULC of the land surrounding each grid square, not the quality of the grid square itself (NDVI). For a given habitat type, Habitat Quality is then based on this definition of Habitat Degradation. Is this interpretation justifiable in the case of snub-nosed monkey habitat? What evidence or theoretical considerations should make us believe that the quality of snub-nosed monkey habitat is determined by the LULC of surrounding lands? This should be tackled explicitly in the Discussion section - more than a single sentence in Lines 273-274.
3. The paper includes an analysis of the impact of the drop-off in impact of surrounding LULC on habitat suitability, with estimation of the misleadingly titled “maximum effective distance” (also termed “distance threshold value,” which I suggest is a better label). This is of particular interest and could be an important contribution. Unfortunately, the description of how this was modeled is particularly opaque, and no justification for the approach is provided. The “correlation analysis” mentioned in Line 170 presumably means that the authors used an exponential regression or GLM of current vegetation cover (as measured with the standard approach of NDVI) on LULC of surrounding grid squares at various distances away from the grid square in question. This should be described in detail.
4. In the InVEST model’s description on the web, the guidelines suggest that the user sets the drop-off (“maximum effective distance”). The original intention of the authors of the InVEST model appears to have been to model the impacts of edge effects, such as introduction of invasive species, microclimate, fire threat, etc. on biodiversity. In this article, however, the focus is on a single species of primate, and the rationale for the calculation of the distance threshold values is far from clear. What is the model of change? How should we imagine LULC outside of a patch affect the current NDVI of the patch? Correlation is not causation.
5. The authors, however, apparently have enough data to examine change in NDVI across time. This should have allowed them to model the relationship between surrounding LULC in one time period and future habitat change into the next time period, as measured by the delta NDVI between the current and next time period. This would have been a much more meaningful analysis, and I suggest that the authors take the time to reanalyze the data with this approach, which would make for a greatly improved argument, and indeed an important publication.
6. The equation for the model at the core of InVEST (on Line 131) includes a term for the impact of the legal status of land, beta subscript x. The authors include this in the equation, but give no information elsewhere on how, or even whether, this was considered. If legal status was considered, a simple table of the values assigned to this Beta factor for each legal classification should be presented (e.g., for nature reserve core zone, buffer zone, experimental zone, forest park, collective forest and other). Better yet, the authors might have been able to use change in NVDI and a Bayesian or GLM approach to derive the values of Beta for each class of legal designation, with confidence intervals. This would have made for a powerful and interesting analysis.
7. Rarity, Lines 141-145: The index of rarity, R subscript j, is generated by a clever subroutine of the InVEST program. This is potentially valuable information for general biodiversity conservation, but difficult to interpret. For example, presumably, the LULC category plantation forest would be counted as a rare habitat when plantations were first introduced, then become less rare over time. But is this meaningful? Or were only natural LULC listed in Table 2 considered for rarity calculations? If so, this should be stated clearly. Even then, is it relevant to snub-nosed monkey conservation? If printed space is at a premium, then this and the sections about rarity in the Results section (Lines 223-234) and Discussion section (Lines 284-291) can be removed.
8. Recommendations, Lines 292-322: These recommendations for conservation action are refreshingly detailed and broad in scope. They all seem appropriate.
In summary, the authors have done a service by introducing the InVEST approach to a new audience but did not deliver on all the model promises. I do believe that the approach is an interesting and important one and wish the authors success as they reanalyze their data in a more robust way.
Minor issues:
Lines 43-44: The habitat of the Yunnan snub-nosed monkey is included in a UNESCO-recognized World Heritage Natural Site. It is reported to have the highest biodiversity of any temperate region in the world.
Line 145: The equation is in error. The summation should be over j, not x. The R subscript j should not be in subscript case.
Line 130, Reference 42: This may not be an appropriate reference. The original source for the equation should be referenced here, which I believe is available at https://storage.googleapis.com/releases.naturalcapitalproject.org/invest-userguide/latest/en/habitat_quality.html
Lines 129-130: D is the “degradation score” referred to above, and this needs to be stated here for clarity. The value of z, the scaling parameter, should be stated here. (I understand that it has been fixed in the InVEST model.)
Line 138: Here and elsewhere, a little explanation would make the description much less opaque. For example, “d subscript rmax is the distance at which the impact of the threat has fallen by 95% to 5% of the full value at source.”
Line 138: Why was the exponential model chosen in all cases? Was this chosen by the modeling or by the authors? If the latter, then there is no need to describe the linear model here and column 4 can be deleted from Table 4. If the former, then the mechanism by which the exponential and linear models were compared and how the former was chosen (presumably by the InVEST app) deserves some explanation.
The English grammar and spelling were fine, but the text, particularly explanations of methods often lack clarity.
Author Response
Point 1: 1. With the exception of a generic description of the InVEST approach, which is largely paraphrased from the website (below), the Methods section provides sparse detail about what was actually done. For example, the key step of choosing parameter values for habitat suitability is described with a single general sentence on Lines 155-156, without reference: “The weight of threat factors and the suitability and sensitivity parameters of land cover types were scored by experts.” The information from this scoring, which is valuable in itself, is presented in Table 2. However, the description leaves too many questions unanswered: Who were these experts? What were the principles behind the scoring? What information and what instructions were they given?
Response 1: Changes have been made to section 2.3. Habitat Quality Assessment to address this.
Point 2: 2. Habitat Degradation: Habitat Degradation here is measured with the index generated by the InVEST model. This rests on the LULC of the land surrounding each grid square, not the quality of the grid square itself (NDVI). For a given habitat type, Habitat Quality is then based on this definition of Habitat Degradation. Is this interpretation justifiable in the case of snub-nosed monkey habitat? What evidence or theoretical considerations should make us believe that the quality of snub-nosed monkey habitat is determined by the LULC of surrounding lands? This should be tackled explicitly in the Discussion section - more than a single sentence in Lines 273-274.
Response 2: Changes have been made to section 4. Discussion to address this.
Point 3: 3. The paper includes an analysis of the impact of the drop-off in impact of surrounding LULC on habitat suitability, with estimation of the misleadingly titled “maximum effective distance” (also termed “distance threshold value,” which I suggest is a better label). This is of particular interest and could be an important contribution. Unfortunately, the description of how this was modeled is particularly opaque, and no justification for the approach is provided. The “correlation analysis” mentioned in Line 170 presumably means that the authors used an exponential regression or GLM of current vegetation cover (as measured with the standard approach of NDVI) on LULC of surrounding grid squares at various distances away from the grid square in question. This should be described in detail.
Response 3: Changes have been made throughout replacing “Maximum Effective Distance of Threats” with “distance threshold value” and equating the two terms in section 1. Introduction. Changes have also been made to section 2.3. Habitat Quality Assessment to address this.
Point 4: 4. In the InVEST model’s description on the web, the guidelines suggest that the user sets the drop-off (“maximum effective distance”). The original intention of the authors of the InVEST model appears to have been to model the impacts of edge effects, such as introduction of invasive species, microclimate, fire threat, etc. on biodiversity. In this article, however, the focus is on a single species of primate, and the rationale for the calculation of the distance threshold values is far from clear. What is the model of change? How should we imagine LULC outside of a patch affect the current NDVI of the patch? Correlation is not causation.
Response 4: Changes have been made to section 2.2. Land Use and Land Cover (LULC) and Normalized Difference Vegetation Index (NDVI) and section 2.3. Habitat Quality Assessment to address this.
Point 5: 5. The authors, however, apparently have enough data to examine change in NDVI across time. This should have allowed them to model the relationship between surrounding LULC in one time period and future habitat change into the next time period, as measured by the delta NDVI between the current and next time period. This would have been a much more meaningful analysis, and I suggest that the authors take the time to reanalyze the data with this approach, which would make for a greatly improved argument, and indeed an important publication.
Response 5: While we agree with the reviewer that this would indeed improve our argument and improve the paper overall, we do not have enough data currently to examine change in NDVI across time. One reason for this is image data is difficult to obtain, requiring a significant amount of field investigation.
Point 6: 6. The equation for the model at the core of InVEST (on Line 131) includes a term for the impact of the legal status of land, beta subscript x. The authors include this in the equation, but give no information elsewhere on how, or even whether, this was considered. If legal status was considered, a simple table of the values assigned to this Beta factor for each legal classification should be presented (e.g., for nature reserve core zone, buffer zone, experimental zone, forest park, collective forest and other). Better yet, the authors might have been able to use change in NVDI and a Bayesian or GLM approach to derive the values of Beta for each class of legal designation, with confidence intervals. This would have made for a powerful and interesting analysis.
Response 6: Changes have been made to section 2.3. Habitat Quality Assessment to address this.
Point 7: 7. Rarity, Lines 141-145: The index of rarity, R subscript j, is generated by a clever subroutine of the InVEST program. This is potentially valuable information for general biodiversity conservation, but difficult to interpret. For example, presumably, the LULC category plantation forest would be counted as a rare habitat when plantations were first introduced, then become less rare over time. But is this meaningful? Or were only natural LULC listed in Table 2 considered for rarity calculations? If so, this should be stated clearly. Even then, is it relevant to snub-nosed monkey conservation? If printed space is at a premium, then this and the sections about rarity in the Results section (Lines 223-234) and Discussion section (Lines 284-291) can be removed.
Response 7: Changes have been made to section 4. Discussion to address this.
Point 8: Minor issues: Lines 43-44: The habitat of the Yunnan snub-nosed monkey is included in a UNESCO-recognized World Heritage Natural Site. It is reported to have the highest biodiversity of any temperate region in the world.
Response 8: Changes have been made to section 1. Introduction to address this.
Point 9: Line 145: The equation is in error. The summation should be over j, not x. The R subscript j should not be in subscript case.
Response 9: Changes have been made to section 2.3. Habitat Quality Assessment to address this.
Point 10: Line 130, Reference 42: This may not be an appropriate reference. The original source for the equation should be referenced here, which I believe is available at https://storage.googleapis.com/releases.naturalcapitalproject.org/invest-userguide/latest/en/habitat_quality.html
Response 10: “Minor issues”, paragraph 3 – Changes have been made to the References section to address this.
Point 11: Lines 129-130: D is the “degradation score” referred to above, and this needs to be stated here for clarity. The value of z, the scaling parameter, should be stated here. (I understand that it has been fixed in the InVEST model.)
Response 11: Changes have been made to section 2.3. Habitat Quality Assessment to address this.
Point 12: Line 138: Here and elsewhere, a little explanation would make the description much less opaque. For example, “d subscript rmax is the distance at which the impact of the threat has fallen by 95% to 5% of the full value at source.”
Response 12: Changes have been made to section 1. Introduction to clarify distance threshold value to address this.
Point 13: Line 138: Why was the exponential model chosen in all cases? Was this chosen by the modeling or by the authors? If the latter, then there is no need to describe the linear model here and column 4 can be deleted from Table 4. If the former, then the mechanism by which the exponential and linear models were compared and how the former was chosen (presumably by the InVEST app) deserves some explanation.
Response 13: Changes have been made to section 2.3. Habitat Quality Assessment to address this.
Point 14: Comments on the Quality of English Language The English grammar and spelling were fine, but the text, particularly explanations of methods often lack clarity.
Response 14: Changes have been made throughout the text to address this.